# Language Models Can Teach Themselves to Program Better

**Patrick Haluptzok**
Microsoft Research
haluptzok@live.com

**Matthew Bowers**[*]
MIT
mlbowers@mit.edu

**Adam Tauman Kalai**
Microsoft Research
adam@kal.ai

## Abstract

Recent Language Models (LMs) achieve breakthrough performance in code generation when trained on human-authored problems, even solving some competitive-programming problems. Self-play has proven useful in games such as Go, and thus it is natural to ask whether LMs can generate their own instructive programming problems to improve their performance. We show that it is possible for an LM to synthesize programming problems and solutions, which are filtered for correctness by a Python interpreter. The LM's performance is then seen to improve when it is fine-tuned on its own synthetic problems and verified solutions; thus the model "improves itself" using the Python interpreter. Problems are specified formally as programming puzzles [Schuster et al., 2021], a code-based problem format where solutions can easily be verified for correctness by execution. In experiments on publicly-available LMs, test accuracy more than doubles. This work demonstrates the potential for code LMs, with an interpreter, to generate instructive problems and improve their own performance.

## 1 Introduction

Recent Language Models (LMs) pre-trained for code generation [Chen et al., 2021; Chowdhery et al., 2022; Li et al., 2022; Austin et al., 2021] produce useful code and even achieve non-trivial performance in human programming competitions [Li et al., 2022]. LMs that solve programming problems may help make algorithmic breakthroughs in computer science, such as factoring large integers or designing faster algorithms for multiplying large matrices (useful in ML). However, LMs are generally trained on human-authored code which contains bugs and inefficiencies that are reproduced by LMs [Chen et al., 2021], with ambiguous specifications usually in English or by example.

Inspired by the AlphaZero's success using self-play in Go [Silver et al., 2018], it is natural to ask whether self-play could be used for learning a programming language such as Python, by which we mean: *Can an LM design its own programming problems to improve its problem-solving ability?* This paper demonstrates how LMs, together with an interpreter, can be used to generate *diverse* datasets of *verified-correct* code problems and solutions, which can then be used to improve the LMs themselves through fine-tuning. These synthetic curricula are not only correct but *instructive* in the sense that the test performance of the LMs increases once fine-tuned on these diverse datasets of synthetic coding problems and solutions. Because programming is a universal aspect of computing, it is important (and also perhaps surprising) to discover that these LMs are capable of generating novel and instructive *problems*, in addition to verified-correct solutions.

In addition to solution correctness, *diversity* is a key desideratum of synthetic problems. One could create a dataset of trillions of addition problems such as `assert 173288 + 291124 == y` but such a dataset would be useless outside of arithmetic. Similarly, one function $f$ could be used to create infinite variations by renaming its variables, but this would only teach variable naming and $f$. One could do the same with more problems and transformations,

---

[*]Work done while at Microsoft Research

but any set of human-authored problems (and variants) is inherently limited by the accuracy and effort of human creators. AI systems have the potential to go beyond templates and superficial changes to generate vast quantities of novel challenges and innovative solutions. Moreover, self-play might be necessary to one day *surpass* human code quality, just as AlphaZero surpassed human Go play.

The first challenge in self-play for code LMs, unlike Go where the win-condition is clearly evaluable, is that the goal in code generation is not obvious. *How should problems be specified?* Programming problems are often described in English and/or examples and evaluated with hidden test cases in programming competitions and code-generation benchmarks such as CodeContests [Li et al., 2022], HumanEval [Chen et al., 2021], and APPS [Hendrycks et al., 2021]. While LMs have in fact been shown to be capable of generating largely-correct English programming problems [Sarsa et al., 2022], human oversight is still required for vetting the descriptions and test cases.

**Self-play using programming puzzles.** Our approach is simple but powerful: rather than using English problem descriptions which are ambiguous and hard to verify, we generate *programming puzzles* [Schuster et al., 2021] and solutions. Programming puzzles have been shown to be useful for evaluating the code generation ability of LMs. Puzzles are illustrated in Figure 1 and formally described in Sec. 2, but here we note some key features of puzzles as a problem representation:

- **Machine verifiable.** Like unit tests, puzzles are code-based, and any solution can be easily machine verified for correctness and efficiency by execution.
- **Expressive.** Puzzles can represent any P or NP problem, which includes both easy and hard problems requiring all major algorithmic tools. Surpassing human performance on puzzles would lead to algorithmic and mathematical breakthroughs.
- **Useful benchmarks.** LMs can solve puzzles, with more powerful LMs solving more puzzles, and puzzle-solving also correlates with coding experience among humans.

In this work, we show that LMs can generate a myriad of instructive programming problems in the form of puzzles. We show that it is possible for an LM to generate puzzles and machine-verified solutions which are, in turn, useful for improving that same LM. In our case, puzzles are written in Python and a Python interpreter is used for verification. Our strategy for generating instructive problems that improve test performance is to prompt the LM to generate problems similar to those in a small training set.

**Results.** We evaluate our approach and measure performance gains on a held-out set of human-authored test puzzles using three GPT-Neo models [Black et al., 2021]. We find that these LMs can synthesize correct code in the form of novel puzzles and solutions that are machine-verified to solve the puzzles within an allotted time.

These models more than double their own accuracy on test puzzles when fine-tuned on their own synthetic datasets. We also generate synthetic code using the Codex API, filtered for correctness and efficiency by the interpreter. While the Codex API does not currently provide fine-tuning, the code it generates proves even more valuable for improving the Neo models. We also perform an ablation study to compare the value of filtering with the Python interpreter. Finally, a diversity analysis suggests that the larger models generate puzzles of greater variety and coverage.

**Contributions.** There are three contributions of our work[1]. First, we introduce a procedure that can generate a diverse set of programming puzzles with solutions that are verified correct and efficient in that they execute within a given time bound. Second, we release datasets of 1M synthetic puzzles and solutions along with the source code for our work[2]. Third, we show that the problems are instructive, namely that the LM that generates the problem can improve its own performance on held-out test problems. Our work opens the door to the further research using self-play to improve code generation and other problems.

---

[1]Preliminary version presented at DL4C 2022 https://openreview.net/forum?id=H8cx0iO-y-9
[2]https://GitHub.com/microsoft/PythonProgrammingPuzzles in ICLR2023 directory

```python
def f(c: int):
    return c + 50000 == 174653

def g():
    return 174653 - 50000

assert f(g())
```

```python
def f(x: str, chars=['Hello', 'there', 'you!'], n=4600):
    return x == x[::-1] and all([x.count(c) == n for c in chars])

def g(chars=['Hello', 'there', 'you!'], n=4600):
    s = "".join([c*n for c in chars])
    return s + s[::-1]

assert f(g())
```

Figure 1: Illustrative puzzles and solutions that were synthesized by the Codex language model: the first is a simple equation; the second requires finding a palindrome (string same forwards and backwards) with exactly n=4600 copies of each of a given list of substrings.

**Related work.**  Data augmentation is not new to code generation as multiple works have synthesized tests for human-authored code [e.g. Li et al., 2022; Roziere et al., 2021]. However, data augmentation for test coverage still relies on human-authored problems and has human errors and blind spots, unlike self-play where an AI system can generate comprehensive problems and verified-correct solutions. Input-output pairs have also been synthesized for program synthesis and code generation [Balog et al., 2017; Shin et al., 2019; Alet et al., 2021; Li et al., 2022], though again those augmentations are similarly unverified and limited in diversity. The analogy to games illuminates the difference between self-play and other approaches, as discussed in further related work (Appendix A). For instance, human-in-the-loop approaches are like learning Go by playing against humans, learning from human data is like learning from human games, and learning from templates or other types of external (non-LM) synthetic data is like learning from static synthesizers rather than self-play.

## 2 BACKGROUND ON PROGRAMMING PUZZLES

A *Programming Puzzle* [Schuster et al., 2021] is specified by a verification function $f(\cdot, x)$ which may have zero or more input arguments $x$. A *solution* to the puzzle is a function $g(x)$ such that $f(g(x), x) = \texttt{True}$. Thus, given an input $x$ the solution $g(x)$ must generate an output that satisfies the verifier $f$ for the particular input $x$. Examples of (synthetic) puzzles generated by our systems are given in Fig. 1. The task of a code synthesizer is to produce the code for a solution function $g$ given the source code for the puzzle $f$ and the inputs $x$.

The open-source P3 dataset of Python Programming Puzzles demonstrates that programming puzzles can capture this wide range of challenges from various domains, from trivial string manipulation to longstanding open problems in algorithms and mathematics. Many problems currently used in the evaluation of Codex, AlphaCode, and PaLM-Coder have been rewritten as puzzles. Furthermore, puzzles include numerous classic algorithms such as Towers of Hanoi. Puzzles circumvent the aforementioned ambiguities of natural-language and hidden test cases, because the validity of puzzles and solutions can be directly verified by simply executing code. Our work uses the P3 puzzles but not their solutions.

Schuster et al. [2021] introduced puzzles and showed how to use LMs to solve them. In particular, they construct a few-shot learning prompt consisting of five examples of puzzles interleaved with their solutions and the puzzle to be solved appended, and provide this as input to the LM to generate a candidate solution. This candidate is readily checked for correctness and efficiency by running a Python interpreter with a given timeout. Multiple attempts are made by sampling $k > 1$ times from the LM, at a fixed temperature.

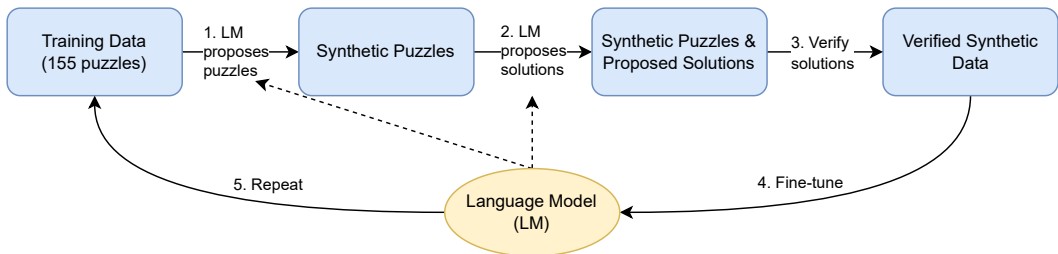

Figure 2: Data Generation Pipeline, used to iteratively generate data and fine-tune the LMs.

## 3 SELF-IMPROVEMENT PIPELINE

The inputs to our system are sets of training (and test) puzzles, five examples of puzzles and solutions used in the the few-shot learning prompt construction, the number $n \geq 1$ of iterations, the number of attempts $a \geq 1$ per puzzle, and the maximum $m \geq 1$ number of synthetic solutions per puzzle. The following four steps are repeated $n$ times:

1. **LM proposes puzzles.** This is done by a few-shot learning strategy: a set of puzzles is randomly sampled from the train set and concatenated together, without solutions. The sequence is chosen at random, without replacement. An example of such a prompt is shown in Fig. 3. The number of puzzles is chosen so that its concatenated length remains within the context window size of the LM, while leaving space for one or more new puzzles to be generated. The language model is then queried, and completes the prompt by generating additional puzzles. The generated puzzles are checked for syntactic validity and also filtered to remove puzzles with "trivial" solutions, such as small constants.

2. **LM proposes solutions.** The valid new puzzles are solved using the "medium prompt" of Schuster et al. [2021] in a few-shot learning approach with a constant number $a$ of attempts per puzzle. This prompt is shown in Fig. 9 in Appendix C.

3. **Verify solutions.** The generated solutions are then verified with the python interpreter. Among the correct solutions, a maximum of $m$ correct solutions per puzzles are selected (if more than $m$ correct solutions are generated, the shortest $m$ are selected).

4. **Fine-tune.** The LM is fine-tuned on this synthetic dataset of filtered puzzle-solution pairs.

Our approach for generating puzzles is motivated by the fact that test and train puzzles are from the same distribution, hence we aim to generate similar synthetic puzzles. While it may be natural to try to generate "hard" problems, they may not be useful if they are from a very different distribution. In each of the $n$ iterations, the LM is used for generation and then the LM is updated by fine-tuning it on the generated data. We next describe our experiments, including how many puzzles were generated, the data, models, constants, and evaluation. We emphasize that no human hand-written solutions are used for fine-tuning or evaluation (other than the five illustrative examples used in the medium prompt for few-shot learning to solve puzzles). An overview of our pipeline for data generation and fine-tuning is depicted in Fig. 2.

## 4 EXPERIMENTS

At the time of writing, the P3 repository contained 397 programming puzzles with 155 puzzles marked as train and 228 as test, as well as Codex-generated solutions to those puzzles. Experiments measure the utility of this process based on how well the LM performs at solving the held-out test puzzles. To this end, we synthesize four datasets of 1 million (1M) puzzle-solution pairs that are verified correct. Each dataset is synthesized using a different LM. The largest model is Codex [Chen et al., 2021] which is accessed via an API. Codex

```
def f(inds: List[int], li=[42, 18, 21, 103, 2, 11], target=[2, 21, 42]):
    i, j, k = inds
    return li[i:j:k] == target

def f(path: List[List[int]], m=8, n=8, target=35):
    def legal_move(m):
        (a, b), (i, j) = m
        return {abs(i - a), abs(j - b)} == {1, 2}
    ...

def f(
```

Figure 3: An example prompt for generating puzzles. For each request for a prompt completion, the LM would generate a new puzzle.

is a GPT-3-like transformer model [Brown et al., 2020] that has been trained on a large corpus of code and a smaller corpus of standalone programming problems. The other three models we generate data from are open-source GPT-Neo 125M, 1.3B and 2.7B models [Black et al., 2021] (henceforth referred to as *Neo*).[3] Neo is a GPT-3-like model which has been pre-trained on the Pile [Gao et al., 2020] dataset including English and GitHub code.

We first describe how we run the pipeline above to generate the four datasets of 1M verified-correct synthetic puzzle-solution pairs. We then evaluate test performance of the Neo models, after being fine-tuned on these datasets. Since fine-tuning Codex is not yet publicly available, we instead fine-tune just the three smaller Neo models on each of these four synthetic datasets and measure test improvements. The baselines are pretrained Neo models.

In two additional experiments, we also evaluate alternative strategies for fine-tuning Neo in our studies. The first is Neo fine-tuned on just the 155 P3 training puzzles with synthetic solutions without any additional synthesized puzzles. Second, we fine-tune Neo on a set of 1M *unverified* synthetic puzzle-solution pairs without correctness filtering. This second baseline enables us to evaluate the effect of automatic correctness filtering.

**Pass@k solving metric.** Consistent with prior work, results are presented using the Pass@k metric [Chen et al., 2021]. Here $k$ is a parameter indicating the number of attempts to solve a puzzle. For each test puzzle, $k$ solutions are generated and the index of the first correct solution obtained for each problem is recorded. Pass@k indicates how many problems had a correct solution generated within the first $k$ solutions. Higher values for $k$ result in solving more problems. We expect Pass@1 performance to improve at lower temperatures, but a single temperature was used to conserve resources. To reduce variance, we in fact generate 256 candidate solutions per puzzle and report results for $k = 1$ to $k = 100$ and use the unbiased estimator of Chen et al. [2021].

### 4.1 Four Datasets of 1M Puzzle-Solution Pairs

Since we did not have the ability to fine-tune the Codex model, the 1M puzzles are all generated in a single iteration $n = 1$ of our pipeline. For the three Neo models, we run only $n = 2$ iterations. The first iteration went slowly as the models produced many invalid puzzle-solutions pairs. In particular, we generated 25K unique puzzle/solution samples from each model in that iteration. However, the fine-tuning greatly increased accuracy and sped up the data generation rate in the second iteration, where we generated 1M unique puzzle/solution samples from each model. This resulted in four datasets of 1M puzzles each, produced by the four different LM's. We refer to these datasets by the model that generated them. After fine-tuning on these 1M new puzzles, we stopped at $n = 2$ iterations as further iterations were costly and the performance increase from iteration 1 to 2 was modest, as can be seen in Figure 7, compared to the generation cost. Possible strategies for generating even more instructive puzzles in later iterations are discussed in Section 5.

---

[3] Neo models were pre-trained by EleutherAI (MIT-licensed), and numbers are parameter counts.

| fine-tune dataset | Verified | Puzzles | Solutions (Count) | # Tokens | Pass@100 |
|---|---|---|---|---|---|
| BASELINE | N/A | No puzzles | No solutions (0) | 0 | 7.5% |
| HUMAN | Yes | Human | Synthetic (635) | 74K | 10.5% |
| VERIFIED-125M | Yes | Synthetic | Synthetic (1M) | 74M | 15.4% |
| VERIFIED-1.3B | Yes | Synthetic | Synthetic (1M) | 65M | 18.9% |
| VERIFIED-2.7B | Yes | Synthetic | Synthetic (1M) | 66M | 20.6% |
| UNVERIFIED-CODEX | No | Synthetic | Synthetic (1M) | 113M | 21.5% |
| VERIFIED-CODEX | Yes | Synthetic | Synthetic (1M) | 98M | 38.2% |

Table 1: Test performance after fine-tuning on the datasets used in our experiments. Pass@100 is shown for 1 epoch of fine-tuning of the Neo-2.7B model on the dataset.

**Puzzle generation.** In order to generate puzzles, we created a simple prompt which consisted of a sample of training puzzles as large as possible for the LM while leaving room for a new puzzle to be generated as illustrated in Fig. 3 (for Codex specifically given the API token limit, this was a median of 43 puzzles). We then applied filtering, eliminating duplicate puzzles, puzzles with an invalid argument type-hint,[4] puzzles which did not parse in Python, and puzzles which had a "trivial" solution, as detailed in Appendix C. For example, if a puzzle took an `int` solution, we tested to ensure that it did not have a solution in $\{-10, -9, \ldots, 100\}$. In total, approximately half of the generated puzzles were eliminated during this pre-filtering process.

**Puzzle solving.** We then used the LM to attempt to solve each of these puzzles, with the same few-shot learning prompt used in P3, which consists of the five tutorial sample puzzles *and solutions* appended with the puzzle to be solved. The exact prompt is shown in Fig. 9. For each puzzle, $a = 128$ attempted solutions were generated by the Neo and Codex models. Each of these candidates was judged as correct or incorrect based on whether it solved the generated puzzle, using the P3 judging code which includes a one-second timeout. We then take the solutions judged to be correct, with up to a maximum of $m = 8$ distinct solutions per puzzle, taking the shortest 8 for the puzzles that had more than 8. Further details including temperatures for generation and solving are in Appendix C.

**Fine-tuning.** Each of the 3 Neo model sizes was fine-tuned for 1 epoch (1 pass through the generated data) using each of the 4 different datasets of 1M synthetic verified puzzle-solution pairs, yielding 12 fine-tuning runs. The format of the fine-tuning data mirrors that of the few-shot solving prompt discussed above and shown in Fig. 9, which is an interleaving of puzzles, solutions, and assertions that the solution is correct for the puzzle. We did not fine-tune the Codex model.

## 4.2 KNOWLEDGE-DISTILLATION ABLATION STUDY

When Codex-generated puzzles are used to fine-tune a Neo model, the smaller model may be learning both from the larger Codex model (a form of what is called *knowledge distillation* [Hinton et al., 2015; Gou et al., 2021]) as well as from the interpreter which filters puzzles for correctness (which might be called *interpreter distillation*). This presented an opportunity for an ablation study to disentangle the effects of the two. To this end, we construct a set of 1M *unverified* synthetic puzzle-solution pairs from the Codex generations *without* correctness filtering. To distinguish these two datasets, we refer to them as UNVERIFIED-CODEX and VERIFIED-CODEX. We fine-tune the Neo models on both of these datasets. This second baseline enables us to evaluate the effect of automatic correctness filtering.

---

[4]A valid puzzle has a single required argument with a type that must be a `bool`, `float`, `int`, `str`, or `List[]`'s thereof, nested to arbitrary depth.

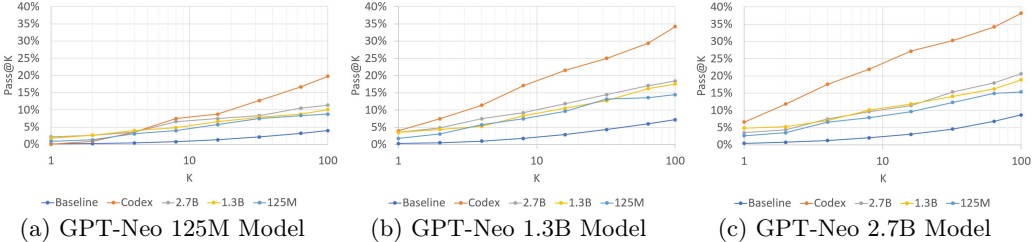

(a) GPT-Neo 125M Model    (b) GPT-Neo 1.3B Model    (c) GPT-Neo 2.7B Model

Figure 4: The graph shows how each model fine-tuned on data generated by the different models (Codex and the three Neo models) impacts Pass@k, with $k$ in log-scale on the horizontal axis. Data generated from the larger models helps more, as larger models appear able to distill more knowledge into the data they generate.

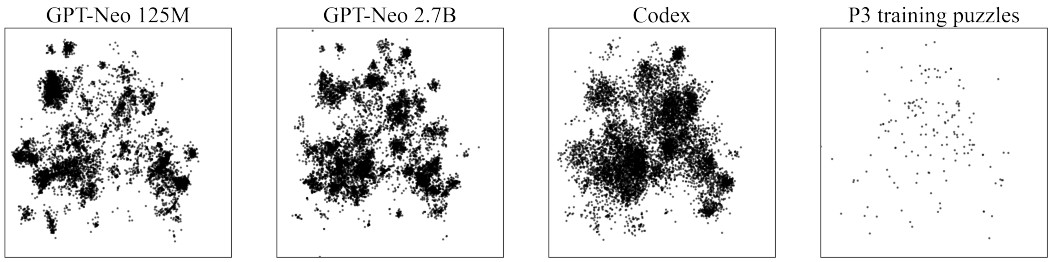

Figure 5: 2D visualization of the puzzles in a sample of 10K puzzles for three of the 1M-puzzle datasets, using Codex embeddings and UMAP dimensionality reduction. Puzzles from larger models have greater coverage (less clumpy) than those of smaller models. Of course, there are many fewer embeddings for the 155 human-authored training puzzles.

### 4.3 RESULTS

We measured how successfully the Neo models solved the 228 test programming puzzles in the few-shot regime (using the same prompt used to solve puzzles during generation), with the Pass@k metric. Each BASELINE model was Neo before fine-tuning. We also considered a HUMAN dataset consisting of correct solutions to 95 puzzles out of the 155 P3 training puzzles. These 635 solutions were generated by a Codex model, as in the original work on Programming Puzzles [Schuster et al., 2021], and verified in the same fashion as described above (Sec. 4.1) for solving the synthetic puzzles. Fine-tuning on that dataset only modestly improved the test accuracy of Neo, presumably because it was so small.

Table 1 shows the Pass@100 performance of Neo-2.7 on all these datasets, as well as the number of tokens in each dataset. Neo, once fine-tuned on any one of the four 1M verified synthetic puzzle-solution pairs, solves 2-5 times as many puzzles as the baseline model, with performance increasing as the model size that generated the data increases. Interestingly, the performance of Neo-2.7B improves even when trained on code generated by Neo-125M because the Neo-125M data has been filtered by a Python interpreter for correctness; effectively Neo-2.7B learns from the "watching its smaller sibling interact with the interpreter."

Fig. 4 shows the significant performance increase in fine-tuning each of the three Neo models on each of the four final 1M datasets. Figure 7 shows a large benefit from the first iteration and small benefit from the second iteration of our pipeline. Due to the small improvement, the process was terminated at $n = 2$ iterations due to cost considerations. In Section 5, we discuss possible directions to improve the results during later iterations. Nonetheless the significant gains from the 25K puzzles are interesting as is the fact that problems generated by larger models seem to be more instructive. The gains, broken down by P3 puzzle domain, are given in Table 3 (page 23). One possible explanation for this is the greater diversity of the puzzles generated by larger models, as analyzed below.

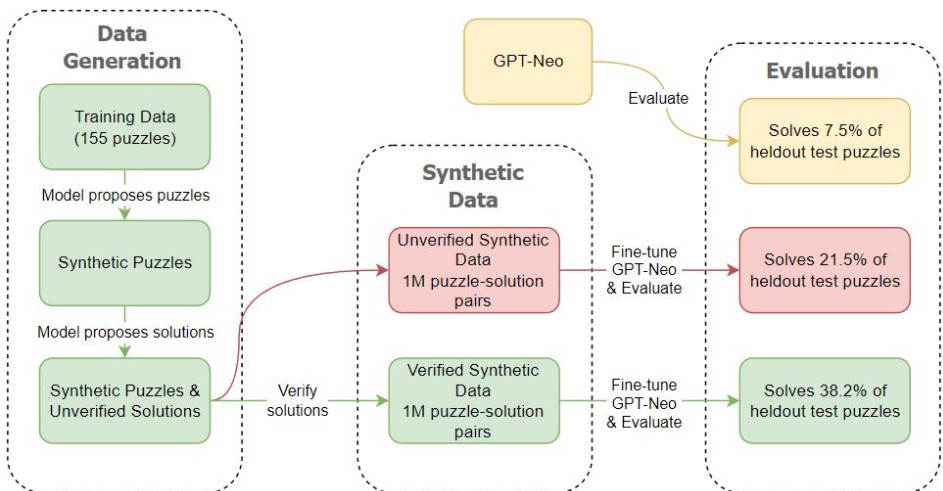

Figure 6: Overview of the Codex ablation experiment and results. Generating and fine-tuning on *verified* synthetic puzzles and solutions, is shown in green, while using *unverified* puzzles is shown in red. The Neo baseline is shown in yellow. All performance results are from the 2.7B model after one epoch of fine-tuning.

**Dataset diversity.** To better understand the diversity present in the synthetic puzzles and how it varies across model size, we use OpenAI Codex API's code embedding feature to generate a 2,048-dimensional embedding of puzzles (not solutions). For each of our four 1M puzzle-solutions datasets, we embedded a sample of 10,000 puzzles. For visualization, we construct 2D embeddings using the UMAP [McInnes, Leland, 2020] dimensionality reduction library (default parameters, densmap=True) on these 40,000 samples in 2048D. UMAP [McInnes et al., 2018] is a dimensionality reduction technique similar to t-SNE [van der Maaten and Hinton, 2008]. Fig. 5 shows these four datasets, with each point being a puzzle. The puzzles from the smaller models appear more densely clustered, while the puzzles from the larger models seem to be more spread out. To quantify this clustering effect, in Appendix D we define an entropy-based diversity metric, given a specific number of clusters $C$, and evaluate it for each of the datasets. As seen in Figure 8 (left), across all numbers of clusters, larger models produce higher entropy (more diverse) results.

**Comparing synthetic puzzles to train and test sets.** In addition to comparing the synthetic puzzles to each other, we also compare them to the training and test puzzles. We measure Euclidean embedding distance between puzzles, which would be 0 for identical puzzles and is inversely related to cosine similarity, since embeddings are unit length. We determine which training and test puzzle are closest to each of the 10,000 sample puzzles for each of the 1M puzzle datasets. Fig. 8 (right) shows the distribution of distances to nearest train versus test puzzle for two datasets. No training puzzles were duplicated and a small fraction of synthetic puzzles were significantly closer to training puzzles than test puzzles, with this discrepancy being more pronounced for puzzles generated by smaller models. Appendix E provides detailed comparison.

**Ablation studies.** The setup and results of the knowledge distillation experiment are summarized in Fig. 6, for Neo-2.7B. The results indicate that a significant part of the performance boost is due to the filtering by the interpreter. The results of Table 1 indicate how much gain is due to differences in model size for the model generating the puzzles. Further details and figures about the ablation are deferred to Appendix G. As shown in Fig. 13 (page 22), fine-tuning on the unverified data improved the Pass@k performance across all models, and verified data gave a considerable boost to performance. Appendix G gives the results of further experiments performed to better understand the results.

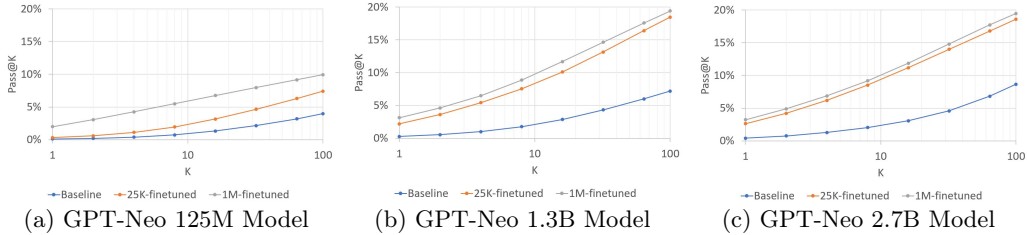

(a) GPT-Neo 125M Model     (b) GPT-Neo 1.3B Model     (c) GPT-Neo 2.7B Model

Figure 7: Pass@k for the three Neo models after fine-tuning on the self-generated and verified data in the 1st iteration (25K samples) and the 2nd iteration (1M samples) of the data generation pipeline. Small gains in iteration 2 suggest that performance may have plateaued.

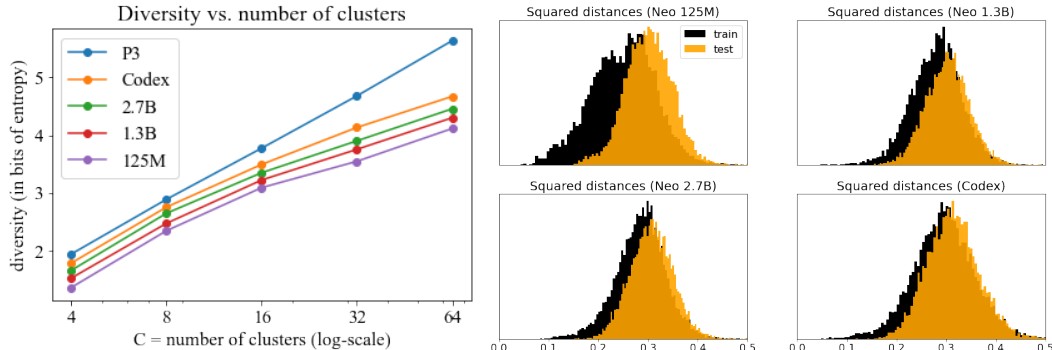

Figure 8: Left: Diversity metrics for the original human-authored P3 dataset and each synthetic 1M dataset, as we vary the number of clusters from 4 to 64 (averaged over 10 runs). Right: Distribution of embedding distance[2] from each puzzle of to their closest puzzles in the training and test sets, across the same four 1M-puzzle datasets.

## 5 FUTURE WORK

Our unsupervised self-improvement pipeline demonstrates that code LMs can improve themselves at solving programming puzzles, using only a Python interpreter and a set of illustrative puzzles (without solutions). LMs synthesize novel puzzles and solutions that lead to self-improvement on test puzzles. It raises several interesting questions for future work.

First, can this approach be improved to the point that LMs generate better code than humans for solving puzzles? Many complex algorithmic challenges can be written as puzzles, e.g., the P3 test set has 150 puzzles derived from the HumanEval benchmark [Chen et al., 2021]. The holy grail here would be solving one of the longstanding open algorithmic or mathematical problems in the P3 dataset, such as the RSA factoring or Collatz puzzles. Future work could use reinforcement learning to improve the generation and filtering stages of the pipeline. While our model's Pass@k performance plateaus when trained on it's own self-generated data, there is no clear limit to what an LM could learn using an interpreter.

An excellent question is how one could use synthetic puzzles to improve code LMs more broadly, e.g., generating code from English descriptions. Transfering gains from one domain to another is a difficult challenge, and simply fine-tuning on millions of synthetic puzzle solutions may make the model "catastrophically forget" [French, 1999] other concepts such as English which are not as useful for solving puzzles. Moreover, benefits may only be realized once the synthetic code is of a higher quality.

The idea of self-play may also be useful in other areas where synthesis and verification can be intertwined, such as theorem-proving. Self-play offers a possible workaround to the data bottleneck for LMs [Hoffmann et al., 2022], since there are significantly larger natural language corpora available for training LMs than source-code repositories. Finally, our self-play pipeline could be combined with other search strategies for code generation.

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

## A    Related Work

Until recently, much work in program synthesis is on Programming by Example (PBE) in Domain-Specific Languages (DSLs), where problems are specified by input-output pairs. This has proven useful in applications such as string manipulation [see, e.g., the survey by Gulwani et al., 2017]. Like English descriptions, PBE is inherently ambiguous. Recent work on massive transformer based LMs [Chen et al., 2021; Schuster et al., 2021; Austin et al., 2021; Li et al., 2022] has enabled synthesis in general-purpose programming languages like Python. Many works have studied data augmentation by synthesizing input-output examples [e.g., Balog et al., 2017; Shin et al., 2019; Alet et al., 2021; Li et al., 2022]. Other works have generated additional tests on top of human-written source code such as Roziere et al. [2021]. Bootstrapping has also been studied in example-based program synthesis [e.g., Menon et al., 2013; Ellis et al., 2021]. However, these works do not consider the AI system itself generating new problems (with verified solutions) as in self-play. LMs such as Codex have been shown to be capable of generating largely-correct English programming problems [Sarsa et al., 2022]. However, human oversight is still required for vetting the descriptions and test cases, and thus their generated datasets are small-scale and contain errors and ambiguities.

To facilitate evaluation, many related datasets of programming problems have been curated, including especially relevant standalone programming challenges described in English and code [Zavershynskyi et al., 2018; Hendrycks et al., 2021; Austin et al., 2021; Chen et al., 2021; Li et al., 2022]. Schuster et al. [2021] and similarly Li et al. [2022] make an important distinction between two types of programming problems: those that only involve *translation* and those that require *problem-solving*. Translation problems, such as "Add up all the odd numbers in array $x$," require the LM to translate a procedure from natural language to code. *Problem-solving* is required when the description does not state *how* to solve the problem. For example, "Find a path of length at most 17 between nodes 1 and 2 in graph $x$" conveys the problem to solve but not how to go about finding a path. Puzzles focus on problem-solving rather than translation.

In knowledge distillation [Hinton et al., 2015] a student model is trained to imitate the behavior of a teacher model on some data, and in the *data-free* paradigm the training data itself is synthetically generated. Related work on knowledge distillation can be found in the survey of Gou et al. [2021]. Recent work in problem solving Cobbe et al. [2021] and commonsense knowledge graphs West et al. [2021] has explored filtering language model outputs for quality during knowledge distillation using a neural filter. This shares the filtering aspect of our work, but given the ambiguity of their natural language task they can't evaluate correctness directly, unlike in the programming puzzle paradigm.

In NLP, various works have considered using data generated by one LM to improve another [Schick and Schütze, 2021; Meng et al., 2022]. Since there is no interpreter to evaluate correctness of natural language, this is more like our ablation knowledge-distillation study than self-play. Ribeiro and Lundberg [2022] use a human-in-the-loop approach to NLP co-generation of datasets, where in some sense the humans can function in place of an interpreter. In the self-play analogy, this would be like humans playing against an AI system as it learns, which still suffers from quality and effort limitations of humans.

In theorem-proving and math-problem solving, [Aygün et al., 2020; Wu et al., 2021; Polu and Sutskever, 2020] show potential value in learning to prove theorems or solve problems from synthetic math problems, though these theorems are not generated by LMs. In the game-play analogy, this is like training an AI system by having it play against other types of bots rather than self-play, and it is unclear whether the goal of beating those bots will lead to improved general performance.

## B    Broader Impact

The automation of writing code may enable software engineers to be more productive and produce higher value products for society. However, increasing software engineer's productivity does risk impacting the total number of software engineers needed, so if substantial gains are made, care would need to be taken when releasing it. Also, automated

software development has serious risks if bugs (e.g., security holes) that are common in the code samples used for training the LM will be reproduced in the LM's output. We refer readers to Chen et al. [2021] and Li et al. [2022] for extensive discussions of the broader implications of code generation.

The approach presented here focuses on teaching the model to solve a problem described in code. Although many natural language problems can be described as a programming puzzle that verifies a solution, some problem descriptions are not so easily translatable into code. Also training exclusively on Programming Puzzles would likely hurt the model's ability to understand natural language. The approach in this paper leverages a deterministic verifier, which isn't available in most problem domains outside code generation, so other approaches like Cobbe et al. [2021] must be used to enable successful filtering for LLM data generation in such domains.

While we do not have access to the data that these models were trained on, given their massive sizes it is possible that they include some Personally Identifiable Information. Despite care taken in their curation, it is also almost certain that they contain offensive content. One symptom of this is the fact that source code of the puzzles we generate contains occasional expletives, not present in P3.

## C  FURTHER DETAILS OF PUZZLE GENERATION AND SOLVING

Fig. 9 shows the prompt used to solve puzzles: the same prompt used (a) in P3 to solve the training puzzles, (b) to solve the generated puzzles, and (c) to solve the test puzzles. It is worth noting that fewer than 1% of puzzles were duplicates. The fixed temperature of 0.9 from prior work [Schuster et al., 2021] was used in all puzzle-solving for generating fine-tuning data, where temperature of 0.8 was used for testing the fine-tuned model per Chen et al. [2021].

In solving puzzles, both synthetic puzzles and P3 puzzles, we use the same judging code from the P3 repository.[5] Their evaluation identifies syntax errors and aborts infinite loops using timeouts. Their judge prevents some malicious instructions from being executed by automated code checks, though other judging systems perform full sand-boxing of the computation to prevent a generated code sample from doing harm like deleting files.

To test of whether a puzzle is trivial or not, we check whether any of the following inputs makes it return `True`.

- For `int` inputs, we test the integers $\{-10, -9, \ldots, 100\}$.
- For `float` inputs, we test `[-100.0, -10.0, -2.0, -1.0, -0.5, -0.1, 0.0, 0.1, 0.5, 1.0, 2.0, 10.0, 100.0]`.
- For `str` inputs, we test `["cat", "dog", "aa", "ab", "foo", "bar", "baz", " "]`.
- For `list` inputs, we test lists of 0-3 items as follows. For lists of `int`, the items are $\{-3, -2, \ldots, 3\}$. For lists of `float`, the items are `[-1.0, -0.1, 0.0, 0.1, 0.5, 1.0, 2.0]`. For lists of `str`, the items are `["a", "b", "foo", "bar", "baz"]`. For lists of `bool`, the items are `True,False`.

All Boolean-input puzzles are deemed trivial because they can be solved by the trivial algorithm that tries both inputs.

## D  FURTHER DIVERSITY ANALYSIS

In this section, we present a detailed diversity analysis. First, Figure 10 shows the embeddings of the puzzles after iteration 1 (a sample of 10K puzzles of the 25K generated puzzle-solution pairs) and the similar embeddings for iteration 2 (a sample of 10K puzzles out of the generated

---

[5]We additionally set the PYTHONHASHSEED environment variable to 0 to make Python `set` functions deterministic.

```python
from typing import List

def f1(s: str):
    return "Hello " + s == "Hello world"

def g1():
    return "world"

assert f1(g1())

def f2(s: str):
    return "Hello " + s[::-1] == "Hello world"

def g2():
    return "world"[::-1]

assert f2(g2())

def f3(x: List[int]):
    return len(x) == 2 and sum(x) == 3

def g3():
    return [1, 2]

assert f3(g3())

def f4(s: List[str]):
    return len(set(s)) == 1000 and all((x.count("a") > x.count("b")) and ('b' in x)
    for x in s)

def g4():
    return ["a"*(i+2)+"b" for i in range(1000)]

assert f4(g4())

def f5(n: int):
    return str(n * n).startswith("123456789")

def g5():
    return int(int("123456789" + "0"*9) ** 0.5) + 1

assert f5(g5())

def f6(inds: List[int], string="Sssuubbstrissiingg"):
    return inds == sorted(inds) and "".join(string[i] for i in inds) == "substring"

def g6(string="Sssuubbstrissiingg"):
```

Figure 9: An example of the prompt used for solving puzzles, identical to the "medium prompt" of P3 [Schuster et al., 2021, Figure C.3]. The first five example puzzles f1-f5 are always the same. The puzzle to be solved is also provided in the prompt as f6, and the solution function signature is provided as g6.

1M puzzle-solution pairs), compared to the human-written puzzles and sample of 10K puzzles from the 1M Codex generated puzzle-solution pairs.

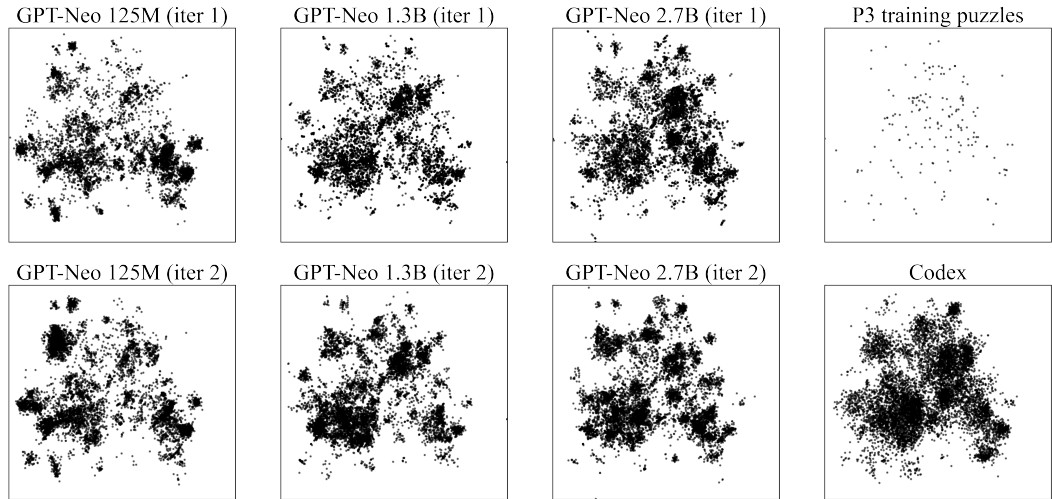

Figure 10: Expanded version of Figure 5 to include the same 2D embeddings of puzzles in a sample of 10K puzzles for the three 25K-puzzle datasets (top, after iteration 1) compared to the three 1M-puzzle datasets (bottom, after iteration 2). Also repeated are the embeddings of the 155 training puzzles and sample of 10K puzzles from the 1M Codex-generated puzzles.

Figure 8 (left) presents an empirical measure of diversity, among the four 1M datasets, in which the puzzles generated by larger models are more diverse. We later illustrate the embeddings of puzzles by showing puzzles from different clusters. Our diversity metric aims to capture the fact that there are many "kinds" of puzzles, and that the distribution over kinds should be diverse within a dataset. The metric depends on a the number of clusters $C$, which we vary, as shown in Fig. 8 (left).

Our diversity metric is computed in two steps. First, we assign each of the puzzles to one of $C$ clusters. To do this, we used K-means clustering (from `scikit-learn` [Pedregosa et al., 2011], default parameters) to cluster the 2048D embeddings of the 397 P3 puzzles (puzzles only—not including solutions) into $C$ clusters. As illustrated below, the clusters appear to be semantically meaningful.

Given any synthetic (or P3) puzzle, we assign it to the cluster whose centroid is closest to the puzzles Codex embedding use the closest of the $C$ cluster centroids to assign a cluster. Sample assignments are also shown at the end of this section.

Once we have assigned puzzles to a cluster center, we compute the distribution over closest cluster center for the 10,000 puzzles in the dataset, call it $p_i$ for cluster center $i$. The total number of puzzles is 10,000 for each dataset (except P3 which only has 397 puzzles). The results are illustrated below for a clustering into $C = 8$ clusters, with random seed 0.

| Dataset | Cl. 1 | Cl. 2 | Cl. 3 | Cl. 4 | Cl. 5 | Cl. 6 | Cl. 7 | Cl. 8 | Entropy |
|---|---|---|---|---|---|---|---|---|---|
| P3 | 9 | 42 | 80 | 60 | 71 | 39 | 48 | 48 | 2.85 |
| Codex | 43 | 1,201 | 1,737 | 763 | 2,918 | 1,390 | 1,063 | 885 | 2.69 |
| Neo-2.7B | 8 | 1,017 | 1,375 | 430 | 3,149 | 1,728 | 864 | 1,429 | 2.60 |
| Neo-1.3B | 1 | 862 | 706 | 303 | 3,019 | 2,748 | 1,763 | 598 | 2.45 |
| Neo-125M | 2 | 1,806 | 354 | 176 | 2,105 | 4,011 | 532 | 1,014 | 2.28 |

The counts can be normalized to be interpreted as a probability distribution over $C$ clusters, with $p_i$ being the fraction of puzzles closest to cluster centroid $i$. The metric is the entropy $\sum_i p_i \log \frac{1}{p_i}$ of this distribution. We report this on the four datasets, as well as the original P3 dataset. Note that if the $K$-means clustering created $C$ identically-sized clusters, then

this metric would be $\log C$ on $P3$, as is reflected in the plot in Fig. 8 (Left). As hypothesized, the larger models generally produce puzzles with greater entropy across all values of $C$, indicating a greater diversity of puzzles and more uniform coverage of the kinds of puzzles in P3.

We now illustrate puzzles from the first cluster of the $C = 8$ clustering above. For each dataset, we show the three puzzles closest to its centroid. In P3, the first two puzzles are from the `human_eval` module.

```python
# P3, 3 puzzles closest to center of cluster 1:
def f(matches: List[int], parens="((())()(()()))(())"):
    for i, (j, c) in enumerate(zip(matches, parens)):
        assert parens[j] != c and matches[j] == i and all(i < matches[k] < j for k
    in range(i + 1, j))
    return len(matches) == len(parens)

def f(matches: List[int], brackets="<<>><<<><<>><<>>>"):
    for i in range(len(brackets)):
        j = matches[i]
        c = brackets[i]
        assert brackets[j] != c and matches[j] == i and all(i < matches[k] < j for k
     in range(i + 1, j))
    return len(matches) == len(brackets)

def f(t: str, s="))(Add)some))parens()to()(balance(()(()(me!)((((("):
    for i in range(len(t) + 1):
        depth = t[:i].count("(") - t[:i].count(")")
        assert depth >= 0
    return depth == 0 and s in t

# Codex, 3 puzzles closest to center of cluster 1:
def f(s: str):
    return any("(" in i and ")" in i and i.count("(") == i.count(")") and not i.
    startswith(")") and not i.endswith("(")
            for i in s.split("()"))

def f(s: str):
    return s.count("(") == s.count(")") and "()" in s and ")(" not in s

def f(brackets: str, pairs='[](<[{)>}]'):
    assert len(brackets) % 2 == 0 and all([i in pairs for i in brackets])
    return brackets == pairs[::-1]

# Neo-2.7B, 3 puzzles closest to center of cluster 1:
def f(s: str):
    return s.count("(") >= 2 and s.count("[") >= 2

def f(s: str):
    return s.count("(") >= 2 and len(s) > 5 or s.count("5") >= 3 and s.count("6") >=
     1

def f(s: str):
    return ((s.count("+") or s.count("-")) or s.count("/") or s.count("*") or s.
    count("\n") == 0) and all(s[i:i+len(s)-1] in s for i in range(len(s)))

# Neo-1.3B, 3 puzzles closest to center of cluster 1:
def f(s: str, i=0, length=5):
    for i in range(5):
        if s[-i] == s[-i + 1] == "".join(s[i:i+length] for i in range(i + length)):
            i += 1
            break
    return len(s) == length # assert length + len(s) == len(s)

def f(s: str):
```

```
    s = s.replace(" ", "").replace("(", "").replace(")", "").replace("]", "").strip
    ()
    return s.count("5") > 0

def f(s: str):
    return "[" in s and s.count("1") == 1

# Neo-125M, 3 puzzles closest to center of cluster 1:
def f(s: str):
    return (s.count("(") - len(")") != len(s) or len(s) >= len(")") and len(s) >= len(
    ") or len(s) + len(") <= len(s)

def f(s: str, chars=['o', 'h', 'e', 'l', ' ', 'w', '!', 'r', 'd']):
    for i in range(len(s) - 1):
        for c in s:
            assert c in s
    return True

def f(s: str, s1="a", s2="b", count=6):
    return s.count(s1) == count or sum(s.count("8") and sum(s) == s2) == 0
```

## E   Further comparison between synthetic and training puzzles

This section provides a more detailed comparison of synthetic and training puzzles. First, we compare the distance between each synthetic puzzle and its closest train and test puzzles. Distances are computed using the aforementioned OpenAI Codex API's code embedding in 2,048 dimensions, on the same sample of 10,000 puzzles for each dataset. Since the API's code embeddings are conveniently unit vectors (length 1), this means that $d^2 = 2(1 - s)$ where $d$ is the distance between two puzzles and $s$ is their cosine similarity, because for any unit vectors $u, v$:

$$\|u - v\|^2 = 2 - 2(u \cdot v) = 2 - 2\cos\bigl(\theta(u, v)\bigr).$$

Thus the findings below reported in distance could equivalently be translated to cosine similarity.

Figure 8 (right) in the body of the paper show histograms depicting the distance distributions in two of the four 1-million puzzle datasets, the smallest and largest Neo models. Histograms for the other models and other iterations were similar. Table 2 shows the averages across these datasets, including both iteration 1 and iteration 2 for the Neo models and the puzzles generated in our unverified Codex dataset. To give a sense of puzzle distances, Figure 12 illustrate with examples of puzzles from the 2.7B dataset and their nearest training puzzles.

Second, Figure 11 compare the length of the puzzles in terms of the number of nodes in their abstract syntax tree. The human generated puzzles tend to be significantly longer. A major effect here is that the unsolved synthetic puzzles were excluded, and longer puzzles are harder to solve. This can be seen clearly in the difference between the Codex verified and unverified

| Model | Avg dist$^2$ train | Avg dist$^2$ test | Difference |
|---|---|---|---|
| Neo 125M | 0.249 | 0.302 | 0.053 |
| Neo 1.3B | 0.280 | 0.303 | 0.022 |
| Neo 2.7B | 0.285 | 0.307 | 0.022 |
| Codex | 0.295 | 0.314 | 0.019 |
| Codex–unverified | 0.298 | 0.313 | 0.015 |

Table 2: The mean squared distance (in embedding space) between puzzles in each dataset and the P3 train and test sets. A small amount of "overfitting" is observed in that puzzles are a bit closer to the training set than the test set, with more overfitting is (greater difference column) observed for smaller puzzles. The numbers in each column are for samples from the 1M puzzle datasets.

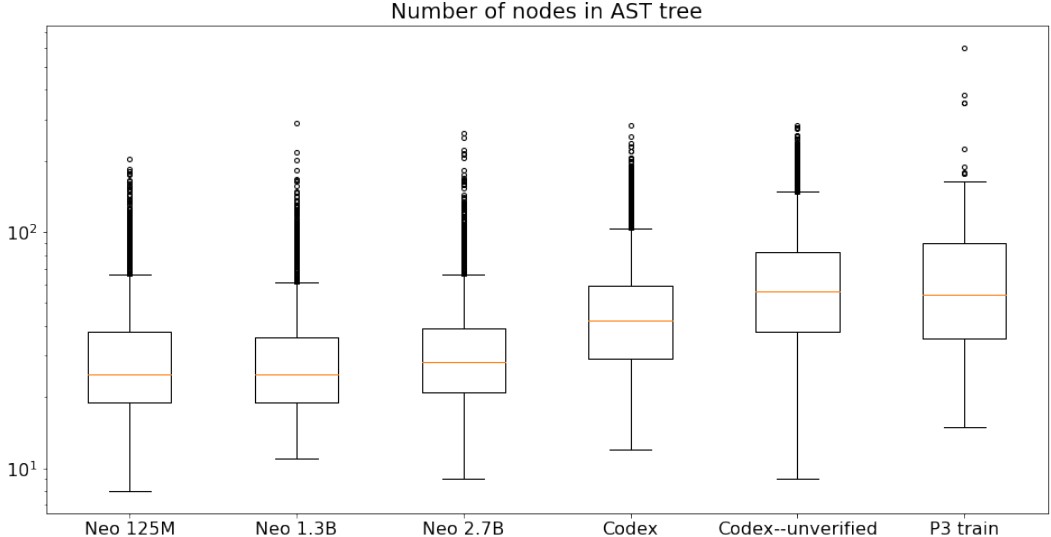

Figure 11: Box plots showing the distribution of puzzle sizes (y-axis, as measured by the number of nodes in the Python AST, log scale), across the 1M puzzle datasets and the training puzzles. The verified synthetic puzzles are shorter because puzzles with no solutions were omitted, and longer puzzles are more likely to have no solutions.

puzzles, with longer puzzles for the unverified puzzles (in which unsolved synthetic puzzles were *not* excluded). Additionally, the Codex sovler is significantly larger and more powerful than the GPT-Neo models, which means that harder/longer puzzles would be filtered less often.

## F    FURTHER EXAMPLES OF GENERATED PUZZLES

A hand examination was performed on a subset of the generated puzzles, where we attempted to understand how the puzzles may originate. We found several concepts repeated from the training, other human concepts such as days of the week, and other puzzles that appear to be derived from programming challenges on the web. We found many human concepts misused, such as the perimeter of a triangle being confused with its side. Additionally input variables were sometimes unused, or puzzles did not test what they appeared like they should test because of certain issues they contained. Finally, comments were sometimes generated of varying quality.

For several puzzles, we attempted to delve deeper to understand the origin for the puzzle. For instance, `f2` from Fig. 1 seems similar in spirit (but not identical) to several of the training problems. Here is a P3 training problem that is somewhat related:

```python
def train(s: str, substrings=['foo', 'bar', 'baz']):
    return all(sub in s and sub[::-1] in s for sub in substrings)
```

Both involve testing palindromes and substrings.

More surprisingly, the following sophisticated problem was generated:

```python
def f(n: int, target=20151120):
    assert 0 <= n <= 1e14
    next = lambda x: (x * 252533) % 33554393
    seen = set()
    now = 20151120
    while now not in seen:
        seen.add(now)
        now = next(now)
```

```python
# Squared distance: 0.049 ——————————————————————————————————

# Synthetic puzzle:
def f(stamps: List[int], target=80, max_stamps=4, options=[10, 32, 8, 50, 30, 41,
    70, 45, 23, 100, 2, 38]):
    for s in stamps:
        assert s in options
    return len(stamps) <= max_stamps and sum(stamps) == target

# Closest training puzzle:
def f(stamps: List[int], target=80, max_stamps=4, options=[10, 32, 8]):
    for s in stamps:
        assert s in options
    return len(stamps) <= max_stamps and sum(stamps) == target

# Squared distance: 0.246 ——————————————————————————————————
# Synthetic puzzle:
def f(s: str, strings=["sand", "water", "fish", "sun"], n=4):
    return s in strings and len(set(s)) == n

# Closest training puzzle:
def f(s: str, strings=['cat', 'dog', 'bird', 'fly', 'moose']):
    return s in strings and sum(t > s for t in strings) == 1

# Squared distance: 0.275 ——————————————————————————————————
# Synthetic puzzle:
def f(s: str, target="dbaabcbbaaacbbaaaca"):
    assert len(s) >= len(target)
    return len(set(s)) == len(set(target)) and all(s[x] == target[x] for x in range(
    len(s)))

# Closest training puzzle:
def f(t: str, s="abbbcabbac", target=7):
    i = 0
    for c in t:
        while c != s[i]:
            i += 1
        i += 1
    return len(t) >= target and all(t[i] != t[i + 1] for i in range(len(t) - 1))

# Squared distance: 0.306 ——————————————————————————————————
# Synthetic puzzle:
def f(string: str):
    for w in ["hey", "cool", "what", "you", "ok", "keep"]:
        if w in string:
            return True
    return False

# Closest training puzzle:
def f(s: str, chars=['o', 'h', 'e', 'l', ' ', 'w', '!', 'r', 'd']):
    for c in chars:
        if c not in s:
            return False
    return True
```

Figure 12: Puzzles from the Codex 1M dataset and their distance to the nearest training example. In this first pair, the only difference is the options input list. That was the closest pair in the entire dataset. Only a 0.003 fraction of the puzzles have squared distance < 0.1.

```
        if now == target:
            return n == 0
        n -= 1
    return False
    next = lambda x: (x * 252533) % 33554393
    now = 20151120
    n = 0
    while next(now) != target:
        n += 1
        now = next(now)
    return n
```

This problem requires computing a discrete log. While the discrete log problem is notoriously difficult and is the basis of numerous cryptography systems, the number is small enough that it can be solved by a simple loop. The P3 dataset does contain a discrete log problem but it is in the test set. While we could not find the exact code above, the problem itself does appear to be equivalent to the English challenge stated on this programming challenge website: https://adventofcode.com/2015/day/25. It is still unclear how exactly the system generated this code.

The following puzzle asks for a list of triangles of *perimeter* 5, but uses the variable name *side*, suggesting that it may not understand the difference between perimeter and side. The puzzle has an additional constraint which is clearly poor programming as it refers to undefined variables a1 and a2. Consequently, solving this requires finding a list of a single triangle of perimeter 5, such as [[2,2,1].

```
def f(ls: List[List[int]], a=24, b=16, c=24, target=None, side=5):
    for a, b, c in ls:
        assert a <= side and b <= side and c <= side and a + b + c == side, "Invalid
     triangle"
    if not target:
        target = ls[-1]

    def legal_move(m):
        (a, b, c), (i, j, k) = m
        return ((a == side or a == b + c) and a == a1 and a != a2) or a == a2 and a
     != a1 and a != b + c

    a1, a2, a3 = target
    moves = list(zip(ls, ls[1:]))
    return all(legal_move(m) for m in moves)

    return [[a,b,c] for a in range(side+1)
                for b in range(side-a+1)
                for c in range(side-a-b+1)
                if a + b + c == side
                and (a == side or a == b + c)]
```

Several puzzles included concepts (like vowels) and specific strings (like the famous pangram below) that appeared in the training data.

```
def f(w: str, z="The quick brown fox jumps over the lazy dog", n=2):
    return w.count("a") + w.count("e") + w.count("i") + w.count("o") + w.count("u")
    == n and w in z and w != z
```

Many puzzles were not particularly interesting such as the two below, which involve finding a string of a given length containing a given substring, and finding a list of 21 numbers between 1-9 that sum to 100.

```
def f(s: str, t="rome", length=14):
    return len(s) == length == len(set(s.upper())) and t.upper() in s.upper()
```

```python
def f(li: List[int]):
    return len(li) == 21 and all(i in li for i in range(1, 10)) and sum(li) == 100
```

Other puzzles involved very human-like strings:

```python
def f(m: str):
    assert m.startswith("Hello, Salif")
    assert "But, but..." in m
    assert m.endswith("You're great!")
    return len(m) == 282

def g():
    return "Hello, Salif. But, but... If a friend ever said hello to me, I wonder
    where are you from? A freaky fellow? Are you from a freaky galaxy?" + \
    " or are you from a freaky universe or a freaky planet? The answer is no: I'm
    megalomaniac!" + \
    " I know because I don't translate meaning. You're great!"
```

Other puzzles involved human concepts such as the day of week which did not appear in the training data:

```python
def f(days: List[str], x="tue", k=3, n=4):
    nums = {"mon": 0, "tue": 1, "wed": 2, "thu": 3, "fri": 4, "sat": 5, "sun": 6}
    numx = nums[x]
    return (len(set(days)) <= k and (n - len(set(days))) * n >= n * (1 + (n - 1) //
    k) and numx <= n // 2 and
            numx != nums[days[n // 2]] and numx > nums[days[0]] and numx < nums[
        days[-1]])   # right half of week is weekdays
        days[:n//2]   # left half of week is weekends
```

The comments that are generated are sometimes useful and sometimes incorrect.

## G   FURTHER EXPERIMENTS ON CODEX DATA

Figures 13, 14, 15, and 16 show further results when fine-tuning GPT-Neo on the Codex-generated data. Fig. 14 compares our results to the Codex model on our test set, after varying number of epochs of fine-tuning. The Davinci model outperforms the much smaller Neo models. Second, one might expect that fine-tuned models could learn in a zero-shot manner. We tested this hypothesis, but, as seen in Fig. 15, the fine-tuned models benefit from few-shot learning. Even after extensive fine-tuning on the puzzle problem format for over 1 billion tokens, the LM still performed better when prompted with the five examples of puzzles/solutions to prime the model. Third, we performed a temperature sweep to test the sensitivity to temperature, as shown in Fig. 16.

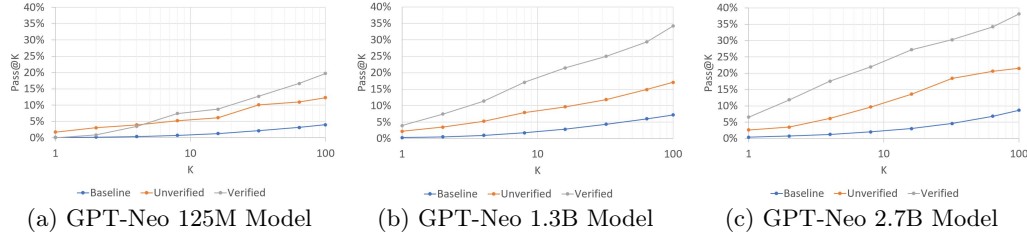

(a) GPT-Neo 125M Model         (b) GPT-Neo 1.3B Model         (c) GPT-Neo 2.7B Model

Figure 13: Pass@k for the three Neo models showing the results of fine-tuning on the unverified and verified data generated by Codex. Data verified correct by the Python interpreter improved accuracy significantly more.

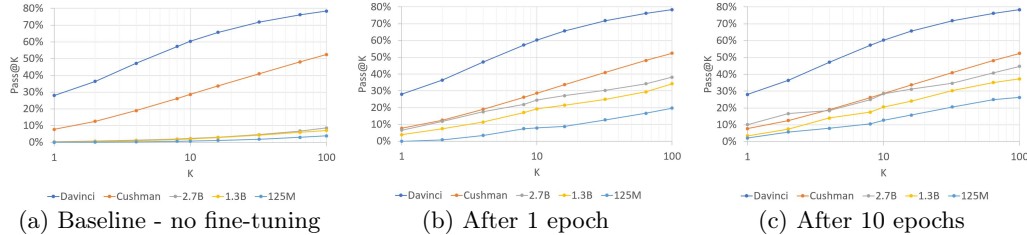

(a) Baseline - no fine-tuning      (b) After 1 epoch      (c) After 10 epochs

Figure 14: Pass@k for the Neo models during fine-tuning, shown in comparison to the Codex models which we were not able to fine-tune (Davinci is 175B, Cushman is 12B in size). Our prompts match the medium prompt style used for baselines in Schuster et al. [2021]. The Neo models were fine-tuned on the 1 million verified puzzles generated by Codex.

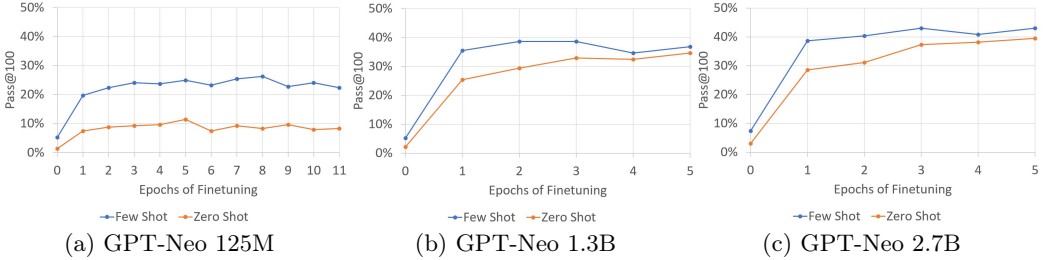

(a) GPT-Neo 125M      (b) GPT-Neo 1.3B      (c) GPT-Neo 2.7B

Figure 15: Few-shot vs. zero-shot and fine-tuning epochs. Across all 3 model sizes, testing Neo in few-shot beats zero-shot significantly even after 11 epochs of fine-tuning which is over 1 billion tokens of Codex-generated puzzle-problem/solution pairs. The LM still benefits from providing the P3 tutorial puzzle prompt.

| Puzzle Type | Puzzle Count | Normalized Baseline | Normalized Finetuned |
|---|---|---|---|
| human eval | 150 | 1.04 | 5.23 |
| user study | 15 | 1.85 | 6.60 |
| trivial inverse | 12 | 1.34 | 8.47 |
| classic | 11 | 0.19 | 0.91 |
| codeforces | 9 | 1.21 | 4.90 |
| (other) | 31 | 0.54 | 2.72 |
| (all) | 228 | **1.00** | **4.95** |

Table 3: In the table above we show the Pass@100 accuracy on the baseline model and the finetuned model for the different domains. The pass rates are normalized by the average Pass@100 for the baseline. The overall test performance improved by a 4.95 factor. The improvement in pass rates from finetuning is the ratio of the two columns.

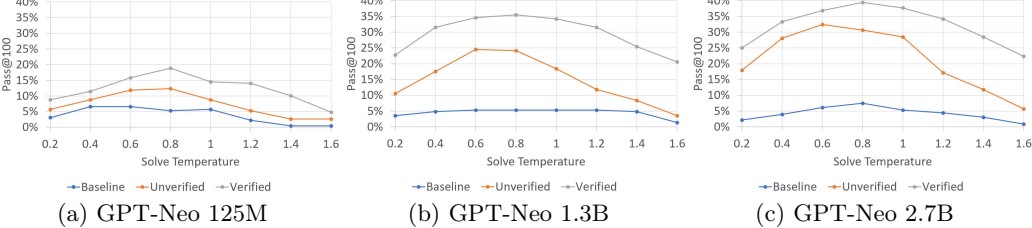

(a) GPT-Neo 125M      (b) GPT-Neo 1.3B      (c) GPT-Neo 2.7B

Figure 16: Temperature controls the amount of diversity in the code solutions generated by Neo. All experiments in our paper were done with a fixed temperature of 0.8, based on the recommendation for Pass@100 in Chen et al. [2021]. A hyper-parameter sweep on temperature across all 3 model sizes verified that 0.8 was also optimal for our model and dataset at a 0.2 search step size for maximizing Pass@100 which is the percentage of problems solved at least once with 100 generated code solutions per problem. Neo was fine-tuned for 1 epoch (≈92 million tokens) in these graphs.

