# OpenReview forum: "Language Models Can Teach Themselves to Program Better"
_ICLR.cc/2023/Conference — ICLR 2023 poster_

### Official Review · Reviewer_cLPb · 2022-10-24

**Confidence:** 4
**Correctness:** 3
**Technical Novelty And Significance:** 3
**Empirical Novelty And Significance:** 3
**Recommendation:** 8

**Clarity, Quality, Novelty And Reproducibility:**

The paper is very clearly written.
I believe the approach is novel.
Since authors plan to release the dataset of synthetic puzzles and solutions and based on the descriptions of their work in the paper, the results should be reproducible.

---------------------------------------------------------------
ICLR review form does not provide a section for comments/questions about the paper. I am going to post them here:
- It would be good to have some discussion of how this work could transfer to program generation from English prompts.
- There is related work on generating buggy programs and fixes for them in the automated program debugging area. I wonder if this work could also generate dataset for buggy/incorrect puzzle solutions and correct puzzle solutions that could be used in automated program debugging. Also I wonder if there is a way to introduce the automated program debugger into the loop to improve either finetuning or prompts for the correct puzzle solution generation. Possible areas to explore in the future work.


**Strength And Weaknesses:**

Strengths:

- A novel approach to improve LM-based program generation by first generating synthetic problems and solutions and then finetuning LMs on the generated problems.
- Highly improved test results
- Ablation studies of model and interpreter distillation

Weaknesses:

- Authors cannot finetune Codex and therefore there are no results for using the authors' approach to improve state of the art results for the problem. See Figure 12 for the gap of the Neo models even with fine tuning vs Codex models. It is possible (likely?) that Codex models will benefit from finetuning using the additional synthetic problems/solutions, but we don't know for sure.

**Summary Of The Paper:**

Authors propose using LMs to generate programming problems and solutions. These synthetic problems and solutions then are used to fine tune LMs for program generation task. This approach leads to improved test accuracy in program generation task.

**Summary Of The Review:**

I believe the work presented here is useful for improving language model ability to generate programs. It seems to be novel, and it achieves marked improvements when used to fine tune LMs. There is a reasonable expectation that this research could improve state of the art results and also be applied in further advancing language model-based program generation. I recommend accepting this paper.

---

> ### Author Response · Authors · 2022-11-15
> **Response to reviewer cLPb**
>
> We thank **Reviewer cLPb** for their time and insights.
>
> > It would be good to have some discussion of how this work could transfer to program generation from English prompts.
>
> Thank you for this suggestion. We have added two paragraphs to the Section 5, future work, about this topic. It will be of interest to readers and may spawn future work. We note that many English descriptions can be converted to puzzles, e.g., the P3 test set we use contains 150 test puzzles that were translated from the `HumanEval` benchmark (where Pass@100 performance gains were a factor of 4 with finetuning), in addition to major open algorithms problems like factoring.
>
>
> > There is related work on generating buggy programs and fixes for them in the automated program debugging area. I wonder if this work could also generate dataset for buggy/incorrect puzzle solutions and correct puzzle solutions that could be used in automated program debugging. Also I wonder if there is a way to introduce the automated program debugger into the loop to improve either finetuning or prompts for the correct puzzle solution generation. Possible areas to explore in the future work.
>
> That would indeed be fascinating. Thank you for the suggestion.
>
> > I believe the work presented here is useful for improving language model ability to generate programs. It seems to be novel, and it achieves marked improvements when used to fine tune LMs. There is a reasonable expectation that this research could improve state of the art results and also be applied in further advancing language model-based program generation. I recommend accepting this paper.
>
> We are glad you share our enthusiasm about self-improving language models for code.

---

> > ### Comment · Reviewer_cLPb · 2022-12-10
> > **Thank you for your responses**
> >
> > I want to thank authors for their responses.

---

### Official Review · Reviewer_bB2P · 2022-10-25

**Confidence:** 5
**Correctness:** 3
**Technical Novelty And Significance:** 3
**Empirical Novelty And Significance:** 3
**Recommendation:** 6

**Clarity, Quality, Novelty And Reproducibility:**

Clarify: The paper is quite well written
Quality: This is an intriguing idea that has a good potential. But more experiments are needed
Novelty: The data generation + filtering has been explored in the Codex paper. The puzzle format is explored in another work. This paper somewhat combines the idea of the two.
Reproducibility: Due to simplicity, I do not doubt that it should be quite reproducible.

**Strength And Weaknesses:**

Pros:
This method has some potential to augment data in order to improve model's general code understanding, generation and reasoning abilities. It is however similar to what Codex paper describes in their data generation process based on filtering with integration tests. However, this work focuses on puzzles.

Questions:
- The format of such puzzles are quite specific. That is, it expects f(.) and g(.) such that f(g(x)) returns True. This format can also be simplified since g() seems to take no argument and returns a value. The simplified format can be 'assert f(' as prompt, then ask the model to generate the actual values. Or is the f(g(x)) format used for something more general? The paper shows a general function f that takes multiple arguments in Figure 3. Are the extra arguments simply ways to take in values?

Cons:
- The experiments can be a bit more thorough. It makes sense that if we finetune on such puzzles, the performance on a held-out set of puzzles will be improved. While it is valid to focus on such puzzle generation as one of the tasks for evaluation, a more important question is whether it translates to general coding abilities.
- A better experiment setup is to perform data bootstrapping; that is, we can generate samples and verify, then finetune on the puzzles, then measure the performance on the general code generation benchmarks such as HumanEval or MBPP.
- What of type guardrails are used to detect that the puzzles are not trivial? For example, if the function looks complicated but due to some reason it always goes to the branch that returns true, then any function g() would solve such puzzle.

**Summary Of The Paper:**

The paper proposes to perform data bootstrapping by using a performant language model generate prompts that specify puzzles and finetune models such data (after filtering for correctness).

**Summary Of The Review:**

Overall, this is a nice method where the performance on the data bootstrapping method is quite expected. For completeness, the paper can push a bit further on studying the general code generation abilities after finetuning of puzzles, as described in the section above.

---

> ### Author Response · Authors · 2022-11-15
> **Response to Reviewer bB2P (Part 1/2)**
>
> We thank **Reviewer bB2P** for their time and thoughtful review.
>
> >  While it is valid to focus on such puzzle generation as one of the tasks for evaluation, a more important question is whether it translates to general coding abilities.
> > A better experiment setup is to perform data bootstrapping; that is, we can generate samples and verify, then finetune on the puzzles, then measure the performance on the general code generation benchmarks such as HumanEval or MBPP.
>
> That is *a* million dollar question, but not the only one. We have added two paragraphs to the future work section to discuss this direction. However, we believe performance gains on puzzles are inherently of interest because:
>
> * Many existing problems can be represented as puzzles. The P3 test set, which we use in our experiments, contains 150 test puzzles that were translated from the `HumanEval` benchmark. We have added a breakdown of improvements by domain, and the Pass@100 accuracy improvements seen on the finetuned model for the `HumanEval` domain are a factor of 4 better than the baseline model, shown in the Appendix in Table 3.
>
> * The P3 dataset includes major open algorithms problems (like factoring puzzles, with hundreds of thousands of dollars in unclaimed prizes). On the road to these breakthroughs, easy-to-medium-difficulty puzzles use all major algorithmic tools, including dynamic programming, recursion, randomized algorithms, etc., requiring LMs to produce correct and efficient code that is *longer* than currently possible.
>
> In our future work discussion, we observe that there transferring improvements from puzzles to other domains is very interesting but may face challenges such as: (1) aggressive fine-tuning on millions of puzzles may make the LMs worse on other things like understanding English, which is sometimes called *catastrophic forgetting*; and (2) the synthetic code may not (yet) be good enough to realize improvements in other domains, compared to existing human GitHub code.
>
> To illustrate how puzzles can capture important problems, RSA offered \$200,000 for factoring a number. It's factorization is still unknown. Here it is in puzzle format:
>
> ```python
> def sat3(i: int, n=2519590847565789349402718324004839...(617 digits)...0357):
>     return 1 < i < n and n % i == 0
> ```
>
> Any improvement on the runtime of factoring algorithms would be considered a major breakthrough. Time limit constraints in evaluating puzzles enforce efficiency.
>
>
> To illustrate how puzzles can capture HumanEval problems, here is an example from `HumanEval`:
>
> ```python
> def intersperse(numbers: List[int], delimeter: int) -> List[int]:
>     """ Insert a number 'delimeter' between every two consecutive elements of input list `numbers'
>     >>> intersperse([], 4)
>     []
>     >>> intersperse([1, 2, 3], 4)
>     [1, 4, 2, 4, 3]
>     """
> ```
> The HumanEval spec for this problem includes 3 hidden test cases:
>
> ```python
> def check(candidate):
>     assert candidate([], 7) == []
>     assert candidate([5, 6, 3, 2], 8) == [5, 8, 6, 8, 3, 8, 2]
>     assert candidate([2, 2, 2], 2) == [2, 2, 2, 2, 2]
> ```
>
>
> Here is the corresponding P3 puzzle in its entirety:
>
> ```python
> def f(li: List[int], nums=[12, 23, -2, 5, 0], sep=4):
>     return li[::2] == nums and li[1::2] == [sep] * (len(nums) - 1)
> ```
>
> Explanation,  `li[::2]` means every other element of the list, so this sublist should match `nums` and `li[1::2]` are the odd elements which must all be separators. This representation, though obscure, is useful for teaching the system to code. There are no hidden test cases. A separate puzzle is needed, however, if one wishes to test more than one instance.

---

> > ### Author Response · Authors · 2022-11-15
> > **Response to Reviewer bB2P (Part 2/2)**
> >
> > > The format of such puzzles are quite specific. That is, it expects f(.) and g(.) such that f(g(x)) returns True. This format can also be simplified since g() seems to take no argument and returns a value. The simplified format can be 'assert f(' as prompt, then ask the model to generate the actual values. Or is the f(g(x)) format used for something more general? The paper shows a general function f that takes multiple arguments in Figure 3. Are the extra arguments simply ways to take in values?
> >
> > Generating a function `g()` as opposed to the answer directly allows the model to generate algorithmic solutions to problems as opposed to simply guessing the answer directly. For example, in the Towers of Hanoi puzzle, the solution `y`, which encodes the list of moves, is *much* longer than the program that generates the solution.
> >
> > As discussed at the start of Section 2, `g(x)` can actually take optional arguments, and in general the goal is the synthesize a function such that `f(g(x), x)` is `True`. In our prompts, the extra inputs `x` are given to both `f` and `g` through the python default argument syntax, which is why it appears simply as `assert f(g())`.  This is a quite general format, as a huge range of problems can be encoded in this way - any problem where a verifier `f(.,.)` is given and the goal is to find a solution `g(.)`.
> >
> > To further illustrate, consider the following example
> >
> > ```python
> > def f(y, x=1234423952):
> >     return (y ** 0.5) == x
> >
> > def g(x=1234423952):
> >     return x * x
> > ```
> >
> > In fact, in the original Programming Puzzles paper, as you suggest, they did consider a zero-shot prompt of the format `assert True == f(` which had much worse performance than the approach of generating `g` because the former requires the solution `1523802493271298304` while the latter requires only `return x * x`
> >
> > As you also observed, the parameter isn't totally necessary. An equivalent variation would be:
> >
> > ```python
> > def f(y):
> >     x=1234423952
> >     return (y ** 0.5) == x
> >
> > def g():
> >     x=1234423952
> >     return x * x
> > ```
> >
> > > What of type guardrails are used to detect that the puzzles are not trivial? For example, if the function looks complicated but due to some reason it always goes to the branch that returns true, then any function g() would solve such puzzle.
> >
> > One step in our pipeline is to remove so-called *trivial* puzzles which have small constant solutions, as you suggest. This is described in the second paragraph of Section 4.1, and details are given in Appendix C.
> >
> > > Overall, this is a nice method where the performance on the data bootstrapping method is quite expected. For completeness, the paper can push a bit further on studying the general code generation abilities after finetuning of puzzles, as described in the section above.
> >
> > We consider the approach a useful step forward in computers generating algorithms, due to the expressivity of puzzles. Eventual benefits for code generation are certainly of interest, but we feel that demonstrating the potential of self-improvement may itself be of value to other scientists working on the topic.

---

### Official Review · Reviewer_qoTN · 2022-10-25

**Confidence:** 4
**Correctness:** 3
**Technical Novelty And Significance:** 2
**Empirical Novelty And Significance:** 2
**Recommendation:** 6

**Clarity, Quality, Novelty And Reproducibility:**

Clarity
  * I think the paper is very well written. The use of illustrative examples makes it easy to follow.

Quality
  * I think the paper is technically sound. However, some important aspects (e.g, program space of $f$) need clarification.

Novelty
  * The overall algorithmic novelty is a bit unclear to me. I think the paper could make this aspect a bit clearer.

Reproducibility
  * There seems to be sufficient algorithmic and experimental detail. I have no major concerns here. Code would have been nice to have.


**Strength And Weaknesses:**

Strengths
  * The paper investigates an interesting and important problem involving LMs. Techniques for improving LM code-gen performance without significant labeling effort will have broad impact.
  * The paper is well written. The main ideas (e.g., prompt setup) of the paper are described carefully with illustrative examples. I enjoyed reading the paper.
  * The main algorithmic ideas are intuitively clear. The paper is technically sound.
  * The experiments demonstrate the utility of the main algorithmic ideas (generation, verificationnn, fine-tuning) on the test set puzzles.

Weaknesses
  - A more detailed characterization of the program space representable by $f$ would be appreciated. It seems like f is a large subset (if not all) of Python but I'm not sure.
  - The paper could better identify the novel algorithmic contributions. For example, is the prompt generation idea novel?
  - While the paper does some analysis of the generated data via embeddings and UMAP, I would have liked to see a more detailed analysis and discussion of the generated puzzles. For example, how similar or different are they to the P3 train and test sets? Any overlaps?
  - Although Figure 7 makes it clear that fine-tuning has plateaued, it is not clear why. A deeper analysis of this in conjunction with the analysis of the generated puzzles would have been nice (but outside the scope of the paper). Overall, this seems like an important line of work as the paper states and leaves for future work.

**Summary Of The Paper:**

The paper proposes a method to improve a language model's (LMs) ability to generate programming puzzles. The primary contribution of the paper is algorithmic. Other contributions include the creation of an open-sourced large set of programming puzzles as well as experiments demonstrating the benefits of self-play (where a LM generates puzzles and then fine-tunes on them for improved test-set performance).

**Summary Of The Review:**

The paper tackles an interesting and important problem. Ideas are mostly clear and experiments show improvements. Some aspects (algorithmic novelty, program space of $f$) need clarification which are reflected in my scores (and open to revision depending on author feedback).

---

> ### Author Response · Authors · 2022-11-15
> **Response to Reviewer qoTN**
>
>
> We thank **Reviewer qoTN** for their time.
>
> > A more detailed characterization of the program space representable by  would be appreciated. It seems like f is a large subset (if not all) of Python but I'm not sure.
>
> **Yes**, as discussed in the third paragraph of Section 4.1, we use the judge sandbox from the P3 repository, which allows arbitrary `Python 3.6.9` code (excluding imports and a few statements such as `open`). That [judge code](https://github.com/microsoft/PythonProgrammingPuzzles/blob/main/solvers/codex/lm_solve/judge.py) is publicly available on the P3 repo. Our code will also be made publicly available for reproducibility purposes.
>
> > The paper could better identify the novel algorithmic contributions. For example, is the prompt generation idea novel?
>
> Our overall algorithm is new as is the problem it solves: generating novel *problems* for self-improving language models of code. This is done using a novel combination of approaches that have been used before. It is not uncommon for machine learning contributions to consist of novel combinations of existing tools, or to introduce a new objective that is optimized using existing techniques. While our overall algorithm is novel, no step individually is entirely new.
>
> Regarding prompting language models to generate problems, perhaps the closest work to ours is the very recent work we reference: (Sarsa et al, 2022) which generates English language programming problems for human students. This is done, like our work, by prompting the language model to generate further problems, although they use a single example problem rather than a random set. They have to vet their problems since they cannot be automatically evaluated.
>
> > While the paper does some analysis of the generated data via embeddings and UMAP, I would have liked to see a more detailed analysis and discussion of the generated puzzles. For example, how similar or different are they to the P3 train and test sets? Any overlaps?
>
> This is an excellent suggestion to improve the paper, thank you! We have added a more detailed analysis. First, we analyze the puzzle similarity to training/test puzzles. Then we analyze the puzzle size.
>
> Very few synthetic puzzles are near duplicates to our training puzzles. Instead, there is a range of similarity to the training puzzles, and a bit less similarity to the test puzzles. We have added an analysis using Codex's code embeddings of the puzzles. We computed mean squared distance to the nearest train/test puzzle embedding. This distance would be 0 for identical puzzles. In a sample of 10,000 synthetic puzzles there were a handful of near-duplicate puzzles that were almost the same as the training puzzles. We have added histograms to the paper (see Figure 8) to show the full train/test distributions, and examples to illustrate the distance (see  Appendix E, which we added due to your suggestion). Here is a table of the average squared distance to the closest train versus test embedding.
>
> | Model | Avg $\text{dist}^2$ train | Avg $\text{dist}^2$ test | Difference |
> | --- | --- | --- | ---  |
> | Neo 125M | 0.222  | 0.292  | 0.070  |
> | Neo 1.3B | 0.270  | 0.301  | 0.030  |
> | Neo 2.7B | 0.270  | 0.302  | 0.032  |
> | Codex | 0.295 | 0.314 | 0.019 |
>
> As you can see, there is a small amount of "overfitting" in that puzzles are closer to train than test, and small models exhibit more overfitting. (Speculation, not in paper: one reason the language model avoids near-duplicates could be because the dozens of puzzles in the prompt are quite different from one another.)
>
> Additionally, the new Figure 11 shows an analysis of average puzzle sizes, as measured by number of nodes in the Abstract Syntax Tree (AST), for the 1M puzzle datasets and also for the hand-written puzzles from P3:
>
> | Model | Avg number of nodes in AST |
> | --- | ---  |
> | Neo 125M | 31.0 |
> | Neo 1.3B | 29.9 |
> | Neo 2.7B | 32.2 |
> | Codex | 47.8 |
> | Codex--unverified | 65.4 |
> | Hand-written from P3 train | 74.6 |
>
> The main reason that the smaller models generate shorter puzzles is that unsolved synthetic puzzles were excluded, and longer puzzles are harder to solve. This can be seen clearly in the difference between the Codex verified and unverified puzzles, with longer puzzles for the unverified puzzles (in which unsolved synthetic puzzles were *not* excluded). Additionally, the Codex solver is significantly larger and more powerful than the GPT-Neo models, which means that harder/longer puzzles would be solved more often.

---

### Official Review · Reviewer_t8tu · 2022-10-26

**Confidence:** 3
**Correctness:** 3
**Technical Novelty And Significance:** 2
**Empirical Novelty And Significance:** 3
**Recommendation:** 5

**Clarity, Quality, Novelty And Reproducibility:**

Clarity: The contributions of the work are clearly stated. The descriptions are easy to follow.

Novelty: This paper explores the concept of 'self-play' with respect to code generation tasks and provide a new direction of leveraging programming puzzles as a way to synthesize new 'instructive' samples.

Reproducibility: Evaluation uses publicly available dataset. The fine-tuned models and 1M datasets used for fine-tuning are not yet available. Details about the prompt used for data generation are presented.

**Strength And Weaknesses:**

Strength

- This work explores an interesting concept of using ‘self-play’-like strategy to improve language models for code generation and leverages existing work on programming puzzles to propose a method to demonstrate the effectiveness of the idea.
- Dataset of 1M synthetic puzzles shared by this work could potentially help future research along this direction.

Weakness

- The data generation pipeline does not specify when to stop generating puzzles or solutions in a single iteration. During evaluation 2 iterations are used for Neo-models and 1 for Codex (due to fine-tuning limitations). The overall approach seems adhoc. How to determine the quality of generated puzzles?
- It would be interesting to see if the performance improvements are tied to specific domains of problem.  This would also reflect on the generalizability of the proposed approach over problem instances.
- “In each of the n iterations, the same LM is used for both generation and fine-tuning” -  is the fine-tuned version replaces the base version in each iterations (i.e., is fine-tuned version always expected to be better in each iteration?)


**Summary Of The Paper:**

This paper argues that programming puzzles and their machine verified solutions generated by a Language Model (LM)  can be used to improve the performance of the same LM. It presents a “data generation pipeline” where puzzles for a training set are used as a prompt for the LM to generate new puzzles that complete the prompt. A filtered set of the generated puzzles are used as parts of another prompt for the LM to generate solutions for those puzzles. The generated puzzle-solution pairs are then verified to create a larger training set. Evaluation results show that GPT Neo models fine-tuned with this larger dataset provide improved results measured using the Pass@k metric.


**Summary Of The Review:**

This work explores a potentially effective idea of improving language models for code. The proposed solution seems effective at improving models performance at python based programming puzzle task. However, the proposed data generation pipeline lacks  some details (e.g., determining number and quality of the generated samples, selection of fine-tuned LM in each iteration ). Also, the evaluation could be more informative if accuracy across the domain of problems are discussed.

---

> ### Author Response · Authors · 2022-11-15
> **Response to reviewer t8tu**
>
> Thank you, **reviewer t8tu**, for your time. We appreciate the positive words in the strengths and summary.
>
> > The data generation pipeline does not specify when to stop generating puzzles or solutions in a single iteration. During evaluation 2 iterations are used for Neo-models and 1 for Codex (due to fine-tuning limitations).
>
> Like most work on large language models, resource constraints prohibit extensive hyperparameter searches. Here are some considerations behind our choice of 1 iteration for codex and 2 for the GPT models. As you are aware, finetuning Codex was not available, but OpenAI does provide free throttled API access so we just ran puzzle/solution generation for about 2 months to get 1 million puzzle/solution pairs. For the GPT-Neo models, the number of synthetic puzzles and solutions generated was limited by generation time.  In the first iteration, generating puzzles was quite slow (because most puzzles were unsolved) so we chose to generate only 25K solutions for each model. The second iteration was much faster so we generated the full 1 million solutions. Only 2 iterations were reported because further Pass@100 improvements were not seen in later iterations.
>
> In case you are curious about the performance for different numbers of puzzles, we include the table below.  It shows how the GPT-Neo125M model trained on it's own self-generated data plateaus in Pass@100 test set accuracy at around 5K samples in the first iteration.  So in addition to the practical constraints of slow generation in the first iteration, there are also limited benefit beyond 5K samples.
>
> | GPT-Neo-125M Self-Finetune Set Size | Test-set Pass@100 |
> | --- | --- |
> | 0 | 3.98 |
> | 100 |	4.10 |
> | 250 |	4.21 |
> | 500 | 4.81 |
> | 1,000 | 6.12 |
> | 2,000 | 7.98 |
> | 3,500 | 8.88 |
> | 5,000 | 10.06 |
> | 10,000 | 10.47 |
> | 15,000 | 10.84 |
> | 20,000 | 10.04 |
> | 25,000 | 10.93 |
> | 50,000 | 11.07 |
> | 100,000 | 10.91 |
>
> > How to determine the quality of generated puzzles?
>
> The main quality check of generated puzzles is that a puzzle has a valid solution that executes successfully, and that the puzzle isn't trivially solved by any solution.  Doing a more rigorous quality check on the puzzles and solutions generated would likely lead to better results.
>
> > The overall approach seems adhoc.
>
> The overall approach is: generate puzzles similar to those of the training set, solve them, filter synthetic puzzles and their solutions for correctness, and use these new puzzles and solutions to improve the language model. We hope our explanations have better justified the choices for our iteration counts and puzzle counts in the iterations.
>
> > It would be interesting to see if the performance improvements are tied to specific domains of problem. This would also reflect on the generalizability of the proposed approach over problem instances.
>
> Thank you for this suggestion. We added a breakdown of the improvements based on the test set "domain" to the paper, which we also show below. The test set from P3 is annotated with domains.  In the table below we show the Pass@100 accuracy on the baseline model and the finetuned model for the different domains.  The scores are normalized by the baseline Pass@100 for the whole test set to highlight which categories are harder or easier, relative to the whole test, before and after finetuning.
>
> | Puzzle Type | Puzzle Count | Baseline Pass@100 | Finetuned Pass@100 |
> | ----------- | ----- | ----- | ----- |
> | human eval | 150 | 1.04 | 5.23 |
> | user study | 15 | 1.85 | 6.60 |
> | trivial inverse | 12 | 1.34 | 8.47 |
> | classic | 11 | 0.19 | 0.91 |
> | codeforces | 9 | 1.21 | 4.90 |
> | (other) | 31 | 0.54 | 2.72 |
> | (all) | 228 | 1.00 | 4.95 |
>
> So the overall improvement was a factor of 4.95 and the improvement on `HumanEval` was `5.23/1.04 = 5.03`.
>
> > “In each of the n iterations, the same LM is used for both generation and fine-tuning” - is the fine-tuned version replaces the base version in each iterations (i.e., is fine-tuned version always expected to be better in each iteration?)
>
> **Yes**, the fine-tuned LM updates the base version in each iteration.
>
> > The fine-tuned models and 1M datasets used for fine-tuning are not yet available.
>
> The puzzle datasets and the code will be made public on GitHub once the paper is published and the URL link included in the final not-anonymized version of the paper. We have uploaded the generated iteration 0 datasets to the supplementary materials for review, but adding any of the models or any of the iteration 1 data files exceeds the 100MB supplementary upload limit.

---

### Author Response · Authors · 2022-11-15
**Paper updates**

In response to excellent suggestions from all the reviewers, we have added the following to the paper:

* More detailed analysis of the synthetic puzzles:

    * Figure 8 which shows the embedding distance from synthetic puzzles to training puzzles. We find that there is a small amount of overfitting, with synthetic puzzles slightly closer to the training puzzles than test puzzles. This effect is less significant for larger models.
    * Figure 12 illustrates some puzzles and their embedding distances.
    * Figure 11 which compares the puzzle sizes, in terms of number of nodes in the abstract syntax tree, across datasets. Larger models generate longer puzzles (mainly because only solved puzzles are included in the dataset and larger models can solve longer puzzles).
* A breakdown of the improvements by puzzle domain as annotated in the P3 dataset, in Table 3.
* A discussion in Section 5, Future work, about transferring improvements to the problem of code generation from English description.

We have also uploaded to the supplementary work a version of the paper highlighting the new changes in yellow, as well as the datasets of puzzles (from the first iteration).

---

> ### Author Response · Authors · 2022-11-22
> **Paper updates**
>
> Reviewers, we hope our changes and responses have addressed your feedback and might result in a better score if you think appropriate. If there are any further changes needed, we would be more than happy to make them for the final version. Thanks again!

---

### Author Response · Authors · 2022-12-07
**Requesting reviewer feedback (by Dec. 12)**

Reviewers (other than cLPb), we would appreciate it if you could take the time to read our rebuttal and provide your response. We have carefully considered your comments, substantially improved the paper as a result, and provided detailed responses. If your main concerns have been addressed, we would be grateful if you could update your score accordingly. We thank *all* the reviewers again for your valuable input.

---

### Decision · Program_Chairs · 2023-01-20

**Decision:**

Accept: poster

**Justification For Why Not Higher Score:**

The proposed problem setting is a bit specific to the programming puzzle setting, and therefore not generally applicable to other settings, making its immediate impact a bit limited.

**Justification For Why Not Lower Score:**

The idea presented in the paper is intuitive, and works with good results.  This paper is worth publishing.

**Metareview: Summary, Strengths And Weaknesses:**

This paper presents an approach to improve language model’s performance on the task of solving programming puzzles, without the need of obtaining more labeled data with significant human cost.  The proposed approach uses language models (LMs) to generate more programming puzzles, verify and filter them, then use LMs to generate solutions and verify them, then fine-tune the LMs on the generated puzzle-solution data to get improved performance.

The paper is well-written, the idea is intuitive, and the empirical results are good, with extensive ablations.  Most reviewers support acceptance.

I would encourage the authors to include some additional results from the discussion with reviewers into the paper, for example the performance vs size of the generated dataset seems like a very informative study that could be included in the main paper / appendix.

Go beyond programming puzzles is also worth thinking about, as multiple reviewers raised this directly or indirectly.

**Note From Pc:**

if the above contains the word "oral" or "spotlight" please see: "oral" presentation means -> notable-top-5% and "spotlight" means -> notable-top-25%. As stated in our emails, we are disassociating presentation type from AC recommendations